# Semantic Segmentation of 3D Medical Images Through a Kaleidoscope: Data from the Osteoarthritis Initiative

**Boyeong Woo**[1]                                                    B.WOO@UQCONNECT.EDU.AU
**Marlon Bran Lorenzana**[1]                                  MARLON.BRAN@UQ.NET.AU
**Craig Engstrom**[2]                                             C.ENGSTROM@UQ.EDU.AU
**William Baresic**[2]                                             W.BARESIC@UQ.NET.AU
**Jurgen Fripp**[1,3]                                              JURGEN.FRIPP@CSIRO.AU
**Stuart Crozier**[1]                                              STUART@ITEE.UQ.EDU.AU
**Shekhar S. Chandra**[1]                                    SHEKHAR.CHANDRA@UQ.EDU.AU

[1] *School of Information Technology and Electrical Engineering, University of Queensland, Australia*

[2] *School of Human Movement and Nutrition Sciences, University of Queensland, Australia*

[3] *Australian eHealth Research Centre, Commonwealth Scientific and Industrial Research Organisation, Australia*

**Editors:** Accepted for publication at MIDL 2023

## Abstract

While there have been many studies on using deep learning for medical image analysis, the lack of manually annotated data remains a challenge in training a deep learning model for segmentation of medical images. This work shows how the kaleidoscope transform (KT) can be applied to a 3D convolutional neural network to improve its generalizability when the training set is extremely small. In this study, the KT was applied to a context aggregation network (CAN) for semantic segmentation of anatomical structures in knee MR images. In the proposed model, KAN3D, the input image is rearranged into a batch of downsampled images (KT) before the convolution operations, and then the voxels are rearranged back to their original positions (inverse KT) after the convolution operations to produce the predicted segmentation mask for the input image. Compared to the CAN3D (without the KT), the KAN3D was able to reduce overfitting without data augmentation while maintaining a fast training and inference time. The paper discusses the observed advantages and disadvantages of KAN3D.

**Keywords:** Kaleidoscope, segmentation, CAN, knee, MRI

## 1. Introduction

In medical imaging, image segmentation is often a mandatory step in extracting quantitative imaging biomarkers such as cartilage thickness. Deep learning methods, such as a convolutional neural network (CNN) like U-Net (Ronneberger et al., 2015), have shown to be promising in biomedical image segmentation. However, manual annotation of tomographic images from techniques such as computed tomography (CT) and magnetic resonance (MR) imaging requires significant expertise, is labor-intensive and is subject to reader variability (Garwood et al., 2020). Despite deep learning being a rapidly evolving area of research in medical image analysis, the lack of manually annotated data remains a major challenge in training a deep learning model.

In addition, full-resolution segmentation of CT or MR images using a fully 3D CNN usually requires a large amount of memory. Therefore, patch-based approaches have often been employed for medical image segmentation. For example, Prasoon et al. (2013) used a triplanar patch-based approach where three CNNs were trained to classify the central voxel as cartilage or background. In Ambellan et al. (2019), a combination of 2D CNN, 3D CNN, and statistical shape models (SSMs) were used for knee MR image segmentation; here, the 2D CNN was used for slice-wise segmentation while the 3D CNN was used for segmentation of subvolumes localized along the bone contours.

Besides the lower memory requirement, a benefit of patch-based approaches also includes that data augmentation may be unnecessary if many different patches can be produced from each 3D image. However, as can be seen from the two examples in the above paragraph (Prasoon et al., 2013; Ambellan et al., 2019), patch-based approaches often require multiple steps for full segmentation of one 3D image because each patch only provides a local image context. On the other hand, volume-based approaches (end-to-end volumetric segmentation) preserve the global image context and require only a single inference. In Dai et al. (2022), for example, a 3D context aggregation network (CAN), which has a relatively low memory requirement (compared to other fully 3D CNNs), was used for end-to-end segmentation of prostate and knee MR images.

In this work, we present the kaleidoscope transform (KT) as an alternative to patch-based approach for 3D segmentation of knee MR images. The "patches" produced by the KT preserve the global image context, so it was expected that this approach can provide the benefits of both patch-based approach and volume-based approach. The main contributions of this work are: (1) extension and application of the kaleidoscope transform (KT), proposed by White et al. (2021), to 3D MR images, and (2) proposing a segmentation model KAN3D, a context aggregation network (CAN) with KT.

## 2. Materials and Methods

### 2.1. Kaleidoscope transform

The kaleidoscope transform (KT), proposed by White et al. (2021), formalises the concept of downsampling and concatenating an image with itself. Given a positive integer downsampling factor $\nu$ and integer smear factor $\sigma$, the $(\nu, \sigma)$-KT converts an image (or array) of any dimension into $\nu$ evenly spaced, downsampled images of itself along each axis, each scaled by a factor of $\sigma$. In Lorenzana et al. (2022), a $(\nu, 1)$-KT with an arbitrary downsampling factor $\nu$ was efficiently achieved via element reordering. The current work employed this simplified version of KT modified for use with a 3D image segmentation model (Figure 1).

For 3D images, a $(\nu, 1)$-KT decomposes an image into $\nu \times \nu \times \nu = \nu^3$ downsampled images. In our segmentation model, the downsampled images would then be arranged as a batch instead of concatenating along each axis. In other words, a 3D image with dimensions $1 \times h \times w \times d$ (1 being the batch dimension) would be "rearranged" into a batch of downsampled images with dimensions $\nu^3 \times (h/\nu) \times (w/\nu) \times (d/\nu)$. For example, (4,1)-KT for a 3D image produces 64 voxel-shifted low-resolution copies of the original image (Figure 1(b)).

An efficient $(\nu, 1)$-KT for 3D images was achieved using the Rearrange operation in the Einops package developed by Rogozhnikov (2022).

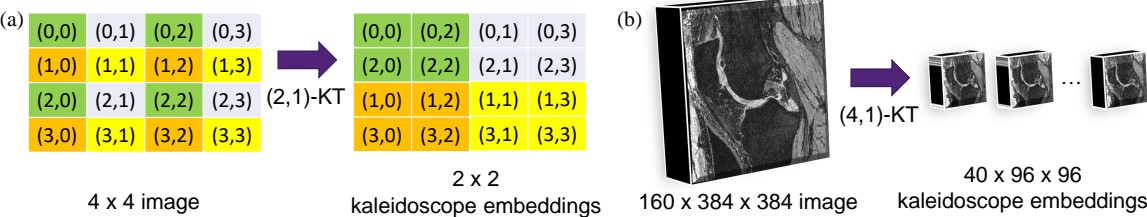

Figure 1: (a) (2,1)-KT for a small 2D image. (b) (4,1)-KT for a 3D image. A 3D input image is decomposed into an array of 64 downsampled images.

## 2.2. KAN3D: Segmentation model

Figure 2 shows our segmentation model KAN3D. The model was based on the 3D version of context aggregation network (CAN) by Dai et al. (2022). The main modification here was the addition of KT before the convolution operations and inverse KT after the convolution operations. This study used (4,1)-KT as shown in Figure 1, to produce a batch of 64 downsampled images. The output from the final convolution layer would then be the predicted segmentation maps for the downsampled images. The inverse KT, which is rearranging all the voxels back to their original positions, would produce the final output—the predicted segmentation mask for the original, full-resolution image.

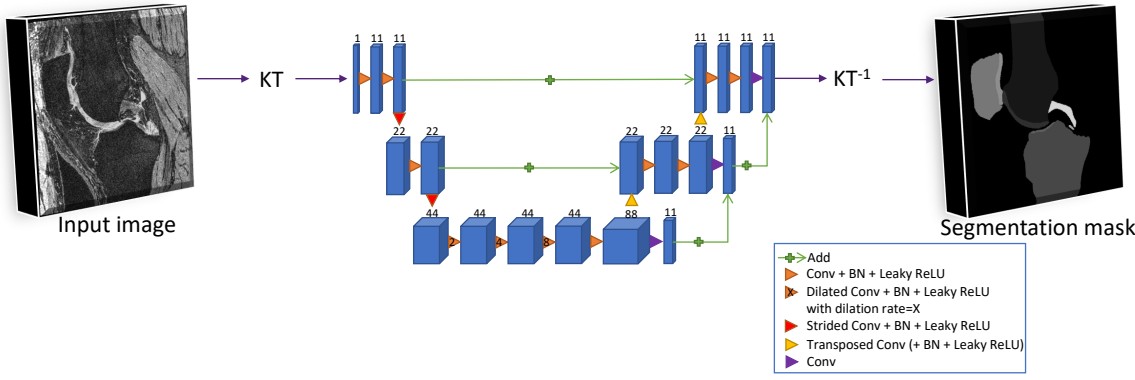

Figure 2: The KAN3D network architecture. KT denotes kaleidoscope transform and KT$^{-1}$ denotes inverse kaleidoscope transform. The bottom-most layers in the figure represents the CAN module. *BN: Batch Normalization*

Ideally, CAN would have no downsampling, but it was difficult to use a reasonable number of filters without any downsampling due to the memory limitation of the graphics card, so two downsampling (and upsampling) blocks were added as a trade-off, along with skip connections (Figure 2). The CAN module is applied after the two downsamplings. To help

stabilize convergence, the network was modified with deep supervision (see Appendix A) by producing secondary segmentation maps at deeper levels of the network and combining them with the final segmentation map via upsampling and element-wise summation.

The multiclass Dice loss function was used for training the network. This loss is often used in medical image segmentation because it intrinsically addresses the class imbalance problem commonly seen with medical images (Milletari et al., 2016; Isensee et al., 2017). See Appendix A for the definition of the loss function and implementation details.

### 2.3. Dataset and experiments

The KAN3D was trained and tested on a small set of knee MR images from the publicly available Osteoarthritis Initiative (OAI) database (Peterfy et al., 2008). Manual segmentations were carried out for a total of 38 MR images consisting of examinations of 25 patients (2 timepoints for some patients). The images are 160 sagittal slices $\times$ 384 $\times$ 384 voxels, and they are all images of the right knee. The MR imaging sequence is 3D DESS (double-echo steady state) with water excitation. Ethics approval for collection of human data was obtained by the OAI and the participating clinical sites (Peterfy et al., 2008).

Segmentation labels consist of the background (0), femoral bone (1), femoral cartilage (2), tibial bone (3), tibial cartilage (4), patellar bone (5), patellar cartilage (6), medial meniscus (7), lateral meniscus (8), posterior cruciate ligament (9), and anterior cruciate ligament (10). The segmentations were carried out manually using ITK-SNAP (Yushkevich et al., 2006) by an analyst (WB) with supervision from another analyst (CE) with expertise in segmentation of the human musculoskeletal system.

The dataset was randomly and evenly split into 5-fold cross-validation sets on a patient basis, meaning there were 30 or 31 images consisting of examinations of 20 patients in each training set and 7 or 8 images consisting of examinations of 5 patients in each test set. Two MR images of the same patient acquired at different timepoints were in the same set.

The KAN3D model was compared to CAN3D, which was the same as the model described in Section 2.2 except that it was without the KT. In addition, the two models, KAN3D and CAN3D, were compared to each other with and without dropouts. In the models with dropouts, dropouts with rate 0.2 were added between convolution blocks during training in an attempt to reduce overfitting. The batch size was 1 for all models although an image was "rearranged" into a batch of 64 downsampled images within the KAN3D through the KT. All models were trained for 100 epochs without any data augmentation.

Segmentation performance was evaluated using Dice similarity coefficient (DSC), average surface distance (ASD), and Hausdorff distance (HD) values. See Appendix B for the definitions of these evaluation metrics.

## 3. Results

Figure 3 shows example outputs from KAN3D and CAN3D trained with or without dropouts. The naïve CAN3D without dropouts produced poor segmentation masks for the bone volumes. The CAN3D with dropouts was better but still erroneous (see Figure 3(c)). The KAN3D with or without dropouts had some errors as well, but their bone segmentation masks were more plausible than CAN3D. Segmentation masks for smaller structures (cartilages, menisci, cruciate ligaments) looked plausible in all models.

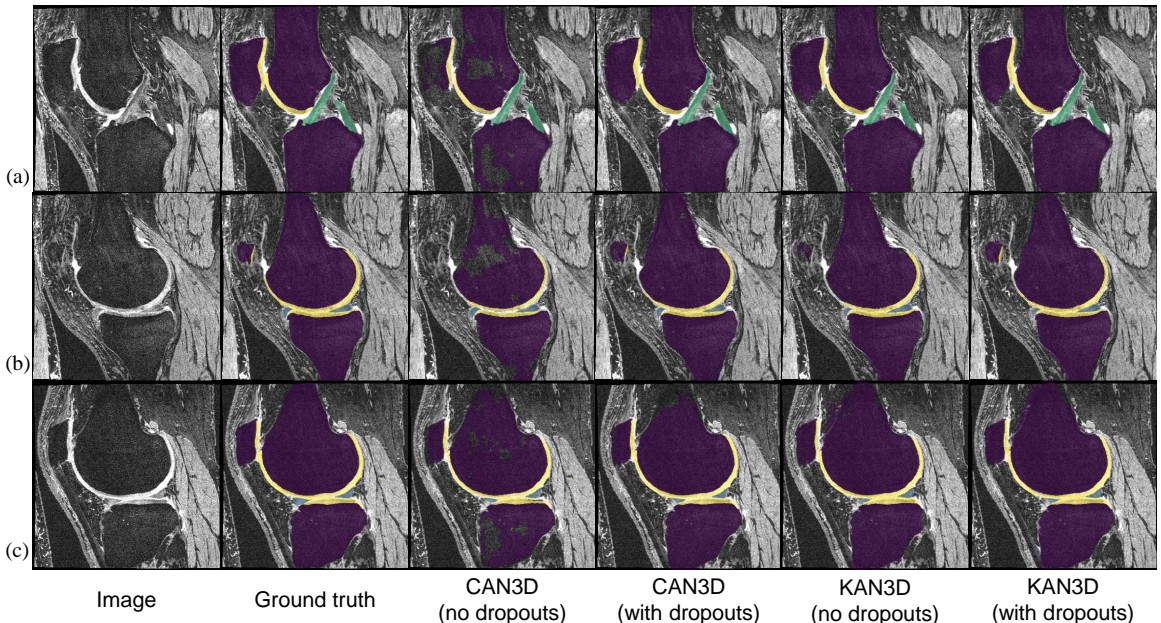

| Image | Ground truth | CAN3D (no dropouts) | CAN3D (with dropouts) | KAN3D (no dropouts) | KAN3D (with dropouts) |

Figure 3: Example outputs from KAN3D and CAN3D trained with or without dropouts, segmenting the anatomical structures in knee MR images from the OAI. In addition to the segmentation of the bones (*purple*) and cartilages (*yellow*) of the femur, tibia, and patella, the slices show the segmentation of (a) anterior and posterior cruciate ligaments (*green*), (b) medial meniscus (*blue*), and (c) lateral meniscus (*blue*).

Table 1 shows the quantitative results. Tukey's honestly significant difference (HSD) test was used to compare all possible pairs of means for each metric. On average, the DSC values were higher with the KAN3D models than with the CAN3D models for the bone volumes, but the converse was true for the smaller structures. Note that because the bone volumes are large, it is relatively easy to reach 90% DSC for the bone labels. The ASD values were lower with the KAN3D models than with the CAN3D models for the bone volumes, but the ASD results were mixed for the other structures. The HD values had a clearer pattern: KAN3D with dropouts produced significantly lower HD values than the other models for all labels.

Dropouts had a relatively small effect in CAN3D. The CAN3D with dropouts even failed to converge for the lateral meniscus label in one of the five cross-validation sets. On the other hand, dropouts significantly reduced average HD values when added to KAN3D.

## 4. Discussion and Conclusion

This study applied the kaleidoscope transform (KT) to a 3D convolutional neural network (CNN) for MR image segmentation. The KT increases the effective receptive field of the

Table 1: Mean DSC, ASD, and HD values for segmentations of the bone, cartilage, meniscus, and cruciate ligament volumes from KAN3D and CAN3D trained with or without dropouts, evaluated using 5-fold cross-validation on 38 MR images (25 patients) from the OAI. Bold represents the best value within each metric.

| Class | Metric | CAN3D (no dropouts) | CAN3D (with dropouts) | KAN3D (no dropouts) | KAN3D (with dropouts) |
|---|---|---|---|---|---|
| **FB** | DSC (%) | 92.2±11.12 | 91.9±13.17 | 95.4±1.37 | **95.4±1.45** |
| | ASD (mm) | 2.18±3.32$^{\ddagger}$ | 1.99±2.68 | 0.94±0.51 | **0.87±0.78** |
| | HD (mm) | 47.43±17.08$^{\ddagger}$ | 44.11±16.44$^{\ddagger}$ | 44.46±20.40$^{\ddagger}$ | **14.12±13.77**$^{\dagger}$ |
| **FC** | DSC (%) | 83.0±4.35 | **84.2±4.86**$^{\ddagger}$ | 82.1±3.49 | 81.6±4.08 |
| | ASD (mm) | 0.39±0.15 | **0.36±0.12** | 0.38±0.08 | 0.37±0.07 |
| | HD (mm) | 24.28±13.25$^{\ddagger}$ | 23.49±14.11$^{\ddagger}$ | 23.12±11.04$^{\ddagger}$ | **10.24±5.08**$^{\dagger}$ |
| **TB** | DSC (%) | 94.4±7.87 | 94.4±9.05 | **96.0±1.64** | 95.9±2.16 |
| | ASD (mm) | 2.29±4.31 | 1.81±4.12 | 0.88±0.95 | **0.74±0.84** |
| | HD (mm) | 60.27±16.16$^{\ddagger}$ | 44.53±26.26$^{\ddagger}$ | 48.84±20.15$^{\ddagger}$ | **20.73±15.61**$^{\dagger}$ |
| **TC** | DSC (%) | 83.1±3.81 | **84.1±4.63** | 82.0±3.41 | 82.9±3.38 |
| | ASD (mm) | 0.39±0.18 | **0.32±0.12** | 0.35±0.07 | 0.34±0.08 |
| | HD (mm) | 26.36±20.98$^{\ddagger}$ | 20.62±21.97$^{\ddagger}$ | 26.00±22.76$^{\ddagger}$ | **7.94±9.66**$^{\dagger}$ |
| **PB** | DSC (%) | 89.4±14.15 | 91.8±6.34 | 92.6±3.26 | **93.3±2.88** |
| | ASD (mm) | 1.13±1.29 | 2.30±3.52$^{\dagger\ddagger}$ | 0.81±0.77 | **0.60±0.39** |
| | HD (mm) | 57.11±29.59$^{\ddagger}$ | 46.27±32.45$^{\ddagger}$ | 53.32±38.56$^{\ddagger}$ | **17.09±27.21**$^{\dagger}$ |
| **PC** | DSC (%) | 78.1±12.28 | **79.6±10.02** | 73.5±10.36 | 77.1±8.54 |
| | ASD (mm) | 0.84±1.10 | 0.60±0.54 | 0.56±0.30 | **0.47±0.22** |
| | HD (mm) | 47.58±29.73$^{\ddagger}$ | 26.06±27.35$^{\ddagger}$ | 34.88±32.04$^{\ddagger}$ | **7.19±11.23**$^{\dagger}$ |
| **MM** | DSC (%) | **77.8±11.27** | 77.7±13.10 | 73.0±12.61 | 73.9±11.62 |
| | ASD (mm) | 0.89±1.13 | 1.08±1.95 | 0.91±1.29 | **0.82±1.17** |
| | HD (mm) | 30.23±22.40$^{\ddagger}$ | 20.85±19.42$^{\ddagger}$ | 25.54±16.88$^{\ddagger}$ | **10.10±10.31**$^{\dagger}$ |
| **LM** | DSC (%) | **80.7±3.81**$^{\dagger}$ | 64.8±34.06$^{*}$ | 77.3±4.06 | 78.6±3.67 |
| | ASD (mm) | 0.64±0.33 | N/A$^{*}$ | 0.56±0.12 | **0.53±0.12** |
| | HD (mm) | 34.73±22.32$^{\dagger\ddagger}$ | N/A$^{*}$ | 24.66±17.98$^{\ddagger}$ | **6.29±5.08**$^{\dagger}$ |
| **PCL** | DSC (%) | 78.7±4.62$^{\dagger\ddagger}$ | **78.9±5.66**$^{\dagger\ddagger}$ | 73.5±5.19 | 74.7±6.90 |
| | ASD (mm) | **0.75±0.27** | 1.17±2.25 | 0.76±0.17 | 0.76±0.23 |
| | HD (mm) | 24.15±20.15$^{\ddagger}$ | 26.21±22.96$^{\dagger\ddagger}$ | 14.14±14.81 | **5.16±4.64** |
| **ACL** | DSC (%) | 73.0±8.66 | **74.0±9.78** | 68.8±10.50 | 71.1±9.07 |
| | ASD (mm) | 0.91±0.54 | **0.70±0.36** | 0.74±0.30 | 0.71±0.27 |
| | HD (mm) | 34.97±19.31$^{\dagger\ddagger}$ | 17.39±19.33$^{\ddagger}$ | 16.77±17.57$^{\ddagger}$ | **6.25±7.60**$^{\dagger}$ |

*FB: femoral bone; FC: femoral cartilage; TB: tibial bone; TC: tibial cartilage; PB: patellar bone; PC: patellar cartilage; MM: medial meniscus; LM: lateral meniscus; PCL: posterior cruciate ligament; ACL: anterior cruciate ligament*

$^{\dagger}$ and $^{\ddagger}$ represent significant difference (p-value $< 0.05$) compared to KAN3D (no dropouts) and KAN3D (with dropouts), respectively, with Tukey's HSD test.

$^{*}$ CAN3D (with dropouts) failed to converge in one of the cross-validation sets, so it was excluded from Tukey's HSD test (for the LM label only).

model and simplifies the overall segmentation problem. As the convolutions are performed on low-resolution volumes, the "distance" between structures decreases and therefore improves volume-awareness of the model. Also, as the number of segmented voxels decreases for each sub-volume, the actual segmentation problem has a reduced dimensionality. In other words, the high-resolution segmentation task is reduced into multiple low-resolution approximations, which are then stitched back together.

Compared to the CAN3D, the KAN3D qualitatively improved the segmentation for the bone volumes (Figure 3), but not for the smaller structures. The reason the KAN3D improved the segmentation of the larger structures is probably that the batch of "down-sampled" images made by rearranging the voxels provides a global context of an image. However, the downside is that the image resolution is decreased, making it more difficult to accurately segment the smaller structures.

The most notable observation was the decrease in HD values with KAN3D with dropouts (Table 1). The very high HD values in the results were mostly caused by random "holes" and "stray voxels" (false negatives/positives) in the segmentation masks. These errors are frequently seen with CNN-based segmentations due to the localized nature of the convolution operations. The KAN3D without dropouts improved the DSC and ASD values for the bone volumes, but not the HD values, compared to the CAN3D. However, the addition of dropouts to KAN3D significantly decreased the HD values for all labels compared to the other models, suggesting that the global attention provided by the KT combined with the regularizing effect provided by the dropouts helps to reduce the large errors.

Figure 4 shows the loss curves for KAN3D and CAN3D trained with or without dropouts. Due to the small size of the training set, the models without dropouts quickly overfitted to the training data. Adding dropouts to CAN3D reduced the degree of overfitting, but a greater amount of reduction in overfitting was observed by adding dropouts to KAN3D. Another notable observation was that the test loss converged in a more stable manner in KAN3D compared to CAN3D in which the test loss tended to fluctuate more. The dropouts seem to have a synergistic effect when used with KAN3D in regularizing the model. This could be because the KT also induces a small amount of regularizing effect similar to data augmentation by producing 64 different (voxel-shifted) low-resolution copies of the original image.

See Table 2 for mean training and inference time per image volume. Despite having additional operations (KT and inverse KT), KAN3D had shorter training and inference time than CAN3D. By decomposing a large 3D image into a batch of smaller images, the KT may have reduced the time required for convolution operations, decreasing the overall time. See also Appendix C for some further discussion.

Table 2: Mean training and inference time per step.

|  | CAN3D (no dropouts) | CAN3D (with dropouts) | KAN3D (no dropouts) | KAN3D (with dropouts) |
|---|---|---|---|---|
| Training speed (s/volume) | 4.3 | 4.3 | 4.2 | 3.9 |
| Inference speed (s/volume) | 2.4 | 2.4 | 2.1 | 1.5 |

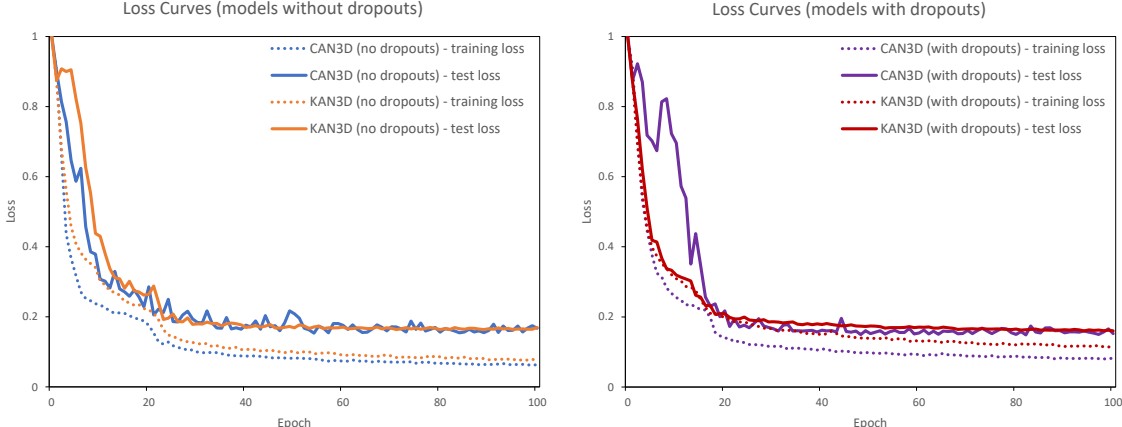

Figure 4: Loss curves showing training loss (*dotted lines*) and test loss (*solid lines*) over epochs for KAN3D and CAN3D trained with or without dropouts. Note that the loss curves shown are for the first of the five cross-validation sets; other sets showed similar pattern.

As described above, the main drawback of KAN3D is the limitation in the segmentation of small structures. Future work could investigate using a hybrid model, e.g. combining the KAN3D with parts of the CAN3D (without the KT), to utilize both local and global context for more accurate segmentation of structures of all sizes. The very small size of the dataset was also a limiting factor in this study. Future study could examine whether KAN3D has any benefits when the size of the training set is larger. Another limitation of the current study is that it applies the KT to CAN3D only. As the KT is a general method that can be applied to any model, one could also look into combining it with another segmentation method such as anomaly-aware segementation (Woo et al., 2022) or nnU-Net (Isensee et al., 2021) to enhance its generalizability. The KT could also be combined with other methods that deal with the problem of small annotated datasets such as semi-supervised learning, data augmentation and self-supervised learning.

In summary, this work demonstrated the benefits and limitations of applying the kaleidoscope transform (KT) to a 3D segmentation model. The proposed KAN3D model has benefits of both patch-based approach and volume-based approach. The study showed that KAN3D with dropouts generalizes well without the need for data augmentation when the training set is extremely small (an advantage of patch-based approach). Yet, since the patches are put back together at the end, it preserves the overall 3D structure and only requires a single inference, having a fast inference time (an advantage of volume-based approach). However, the current study also demonstrated that the model still has its limitations, including limited accuracy in the segmentation of small structures. Future studies are expected to include adding further improvements to the model and minimizing its weaknesses.

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

## Appendix A. Implementation Details

All models (CAN3D and KAN3D) were based on the 3D version of context aggregation network (CAN) by Dai et al. (2022). The original CAN (2D version) was proposed by Yu and Koltun (2016) as an alternative to autoencoder-based CNNs such as U-Net (Ronneberger et al., 2015). In U-Net, the progressive downsampling achieves the effect of integrating contextual information at multi-scale, and the lost resolution is recovered through upsampling. However, since semantic segmentation task requires full-resolution output, there is the question of whether such downsampling and upsampling are truly necessary, which is why Yu and Koltun (2016) proposed using dilated convolutions rather than downsampling for multi-scale context aggregation.

Figure 2 shows our segmentation model KAN3D. Ideally, CAN would have no downsampling, but it was difficult to use a reasonable number of filters without any downsampling due to the memory limitation of the graphics card, so two downsampling (and upsampling) blocks were added as a trade-off, along with skip connections. The CAN module is applied after the two downsamplings. The CAN module consits of convolution blocks with progressively increasing dilation rates and then a non-dilated convolution block as the final block of the CAN module. While non-dilated convolution layers were initialized using the Glorot uniform initializer (Glorot and Bengio, 2010), dilated convolutions were initialized with the identity initializer which was found to be more effective for context aggregation (Yu and Koltun, 2016).

Several previous works on volumetric segmentation using 3D CNNs (typically U-Net-like) have employed a technique referred to as "deep supervision" (Kayalibay et al., 2017; Isensee et al., 2017; Raj et al., 2018), providing integrated direct supervision to the hidden layers, rather than providing supervision only at the output layer (Lee et al., 2015). Deep supervision helps stabilize and speed up convergence by encouraging deeper layers to produce improved segmentation results (Kayalibay et al., 2017). Here, the networks (CAN3D and KAN3D) were modified with deep supervision by producing secondary segmentation maps at deeper levels of the network and combining them with the final segmentation map via upsampling and element-wise summation (Figure 2).

The multiclass Dice loss function was used for training the network. This loss is often used in medical image segmentation because it intrinsically addresses the class imbalance problem commonly seen with medical images (Milletari et al., 2016; Isensee et al., 2017). The loss function is defined as:

$$\mathcal{L}_{DSC} = 1.0 - \frac{2}{|K|} \sum_{k \in K} \frac{\sum_i u_{i,k} v_{i,k}}{\sum_i u_{i,k} + \sum_i v_{i,k}}. \tag{1}$$

Here, $u$ is the softmax output of the network and $v$ is the one-hot encoded ground truth segmentation map; $K$ is the number of classes, and $u_{i,k}$ and $v_{i,k}$ denote the softmax output and ground truth label, respectively, for class $k$ at voxel $i$.

The models were implemented using Tensorflow (Abadi et al., 2015) version 2.4 with Keras API (http://tensorflow.org/guide/keras) and were trained on a high-performance computer with NVIDIA Tesla V100-SXM2-32GB. The "mixed precision" policy in Keras API was used to overcome memory limitation when training 3D CNNs with a large number of feature maps. Mixed precision refers the use of 16-bit floating-point type in parts of the model during training to make it use less memory.

## Appendix B. Evaluation Metrics

Segmentation performance was evaluated using Dice similarity coefficient (DSC), average surface distance (ASD) and Hausdorff distance (HD) (also called maximum surface distance). These are defined as:

$$DSC = \frac{2|B \cap A|}{|B| + |A|}, \tag{2}$$

$$ASD = \frac{1}{|\partial(A)| + |\partial(B)|} \left( \sum_{a \in \partial(A)} \min_{b \in \partial(B)} ||a - b||_2 + \sum_{b \in \partial(B)} \min_{a \in \partial(A)} ||b - a||_2 \right), \tag{3}$$

$$HD = \max \left( \max_{a \in \partial(A)} \min_{b \in \partial(B)} ||a - b||_2, \max_{b \in \partial(B)} \min_{a \in \partial(A)} ||b - a||_2 \right). \tag{4}$$

Here, $A$ and $B$ denote the set of positive voxels in the ground truth segmentation map and the predicted segmentation map, respectively, and $\partial(\cdot)$ denotes the boundary of the segmentation set.

## Appendix C. Further Discussion

The kaleidoscope transform (KT) preserves the global structure within all KT sub-volumes, just at a lower resolution. Therefore, the important distinction between patch and KT is that for KT, continuity between low-frequency objects (i.e. large structures like bone) are still meaningfully represented within all KT sub-volumes. On the other hand, patch-based approaches can still meaningfully represent high-frequency small structures, at the cost of larger image features. This effect can be seen in the KAN3D scores (Table 1), where segmentation of smaller structures does not improve compared to CAN3D.

In addition, in KAN3D, the patches are put back together after the convolutional layers so that the loss function (Dice loss) during training would be calculated based on the original whole image rather than based on individual patches. In other words, the "true volume" and "predicted volume" used for calculation of DSC would be from the segmentation masks on the original image volume rather than individual patch-based masks. This is one of the major differences between KAN3D and traditional patch-based approaches, and this may also help the model to preserve the overall 3D structure.

Another point worth reiterating is that the KT is basically a rearrangement of voxels, so the number of voxels in each batch was the same in both models—i.e. in CAN3D, the input image volume dimension was 1 x 160 x 384 x 384 ($b \times h \times w \times d$) while in KAN3D, it was 64 x 40 x 96 x 96, so the absolute amount of information in each batch was the same in both models. Note that total training and inference time for KAN3D was actually shorter than

CAN3D in the current experiment (Table 2). However, increasing the batch dimension in either model would cause the hardware to run out of memory (because the total number of voxels in each batch is quite high).

One can conceive that the KT can be utilized a bit differently to reduce memory usage instead of time. For example, instead of passing in the "downsampled" images (or "global patches") created by KT all at once as a single batch, one could just pass in a different "version" of the image at each epoch (sort of like data augmentation) since each patch is slightly different at a voxel level. This will serve the purpose of reducing the memory requirement while still preserving all the available information.

