# OpenReview forum: "Semantic Segmentation of 3D Medical Images Through a Kaleidoscope: Data from the Osteoarthritis Initiative"
_MIDL.io/2023/Conference — MIDL 2023 Poster_

### Official Review · Reviewer_zEue · 2023-02-01

**Confidence:** 4
**Preliminary Rating:** 4
**Recommendation:** Poster

**Summary:**

This paper proposes to combine two previous works the 3D context aggregation network  (CAN) and the kaleidoscope transform (KT) into a single framework. The paper claims that, by being patch-based, KT is more efficient than having to deal with a complete 3D volume, while maintaining global consistency.

The work is evaluated on a public database of knee cartilage segmentation and it is compared with the original CAN formulation. An ablation study to see the effect of dropout is also presented. The results show overall increased performance with reduced inference times.

**Strengths:**

* Overall well written
* The proposed framework shows significant speed improvements while demonstrating a good performance that seems to surpass the original CAN
* The idea of the KT, although previously published, is interesting. This work brings a new concept into the domain.

**Weaknesses:**

* The paper consists of combining two previously proposed method: the kaleidoscope trasformer (KT) withing the context aggregation network. As such, the novelty is limited
* To be self-contained, the paper should provide some brief introduction to context aggregation network (CAN) as this is a relatively recent work, which not every reader may be aware of. Section 2.2 mostly describes how the KT is integrated into CAN, but not how CAN works.
* The work makes general claims about the advantages of using the kaleidoscope transform in 3D segmentation models. However, the validation is limited to its use in a specific framework (CAN) for a particular dataset (OAI). It is therefore advised to lower the tone of the claims.



**Deanonymize Review:**

no

**Detailed Comments:**

- The term OAI is only referred to and defined in page 4. This is not a term of common use. Therefore, the title should use the complete set of words. Alternatively it can be removed
- "as can be seen from the two examples above" -> There are no examples above
- The reference for the multi-class Dice loss should be [1].

[1] Milletari, N. Navab, and S.-A. Ahmadi, “V-net: Fully convolutional neural networks for volumetric medical image segmentation,” in International Conference on 3D Vision. IEEE, 2016, pp. 565–571

**Justification of final score**
Although the paper has some weaknesses that could be improved, I think it brings some new ideas that could be worth discussion during the conference.

**Paper Type:**

methodological development

**Questions To Address In The Rebuttal:**

* A problem of patch based approaches is that they may affect the continuity of the segmented objects. How does KT addresses this?
* The results reported in this work for CAN3D are much lower than those reported in the original publication, despite working on the same dataset. Please justify the gap

---

### Official Review · Reviewer_TCXX · 2023-02-02

**Confidence:** 4
**Preliminary Rating:** 4

**Summary:**

The authors of this paper provide a method to segment knee Magnetic Resonance Imaging scans using kaleidoscope transformation in combination with context aggregation networks. They train and evaluate the proposed KAN3D method on a single but large database, namely the Osteoarthritis Initiative (OAI) database.

**Strengths:**

- The clinical motivation is clear and easy to understand.
- The paper, in general, is well written and easy to read.
- While the overall idea is pretty interesting but the results and evaluation is not convincing enough.


**Weaknesses:**

- Figure 1 is not really beneficial since it only shows downsampled patches → This figure should be improved
- It’s not clear how the inverse KD is actually working, especially given the wording “... which is rearranging all the voxels back to their original positions, …” → but how exactly ?
- The authors propose a lightweight method that should compete against patch-based approaches but they have to use downsampling steps so everything fits onto the graphics card ?
- The authors trained only on right-knee MR images. It would be interesting to see the generalization of their method when applied on a left knee MR image during inference.
- “The dataset was randomly and evenly split into 5-fold cross-validation sets on a patient basis, meaning there were 30 or 31 images consisting of examinations of 20 patients in each training set and 7 or 8 images consisting of examinations of 5 patients in each test set” → Not really clear what the different folds contain at the end.
- Why only Dropout of 0.2 → missing some explanation/ablation. At least a sentence why exactly 0.2 and not 0.23 or anything else.
- Using a batch size of 1 for the CAN3D compared to a batch size of 64 for KAN3D does not seem to be a fair comparison even though the 64 “patches” are from the same image since they are 64 low-resolution copies of the high-res image.. → It should be expected that KAN3D will then be more robust than CAN3D when this setup was used..
- Table 1 is way too difficult to understand → either lighten the Table by making it more reader friendly or it should be replaced with a figure. DSC and ASD results are not really discussed in the paper.
According to Table 1, the proposed KAN3D dropout method outperforms all other methods in only 2 out of 10 classes. Since the paper is focused on semantic segmentation one would expect a different outcome.
- Analysis of Table 1 is somehow mainly focused on HD but not on the DSC even though the title starts with “Semantic Segmentation…”.
- “The shorter training and inference time with KAN3D was counter-intuitive since the model has additional operations (KT and inverse KT).” → Why is it counter-intuitive ? There are two additional operations, yes, but KT reduces the resolution of the image drastically (by factor 4 among (x, y, z) axis which makes the pass through the network way faster compared to the full resolution as in CAN3D ? → “However, by decomposing a large 3D image into a batch of smaller images, the KT may have reduced the time required for convolution operations, decreasing the overall time.” → Giving a proper explanation like this, I would not say the training or inference time is counter-intuitive.
- Figure 4: It’s interesting that the loss curve of the CAN3D (no dropouts) for training looks better than the one of KAN3D, why is that?. Additionally, I don’t see/understand the benefit of a test loss in this particular figure.
- Reproducibility issues: Not a reference/sentence that the source code with the implementation will be made public.


**Deanonymize Review:**

no

**Detailed Comments:**

- The authors should evaluate the robustness of their method when applied on left knee MR images as they only trained on images capturing the right knee
- The setup of their evaluation does not seem to be just as the KAN3D methods use 64 low-resolution copies of the high-resolution image and the CAN3D only the high-resolution image (once)
- The authors should find a better way to show the results in Table 1 and should focus more on the DSC scores as they are actually worse for KAN3D in the majority of the cases which is not ideal given the focus of the paper
- Figure 1 could be more enhanced as it nearly brings no benefit to the paper as every reader should know how a downsampled image looks like


**Paper Type:**

both

**Questions To Address In The Rebuttal:**

- How robust and generalized is KAN3D ? What would you expect when the network is used for left knee MR images although it has only been trained on right knee images? Would it be able to create meaningful segmentations or would this be a case of OOD ?
- How is the evaluation just if the KAN3D uses 64 low-resolution copies of the high-resolution image and the CAN3D only the high-resolution image once ?
- How do you evaluate your method based on the DSC score since KAN3D is outperformed in most of the cases by the CAN3D method ?
- Out of curiosity: Since KAN3D is only a 3D model, how would you combine it with the nnU-Net? nnU-Net e.g. uses 3D full-resolution patches. Would you then use 64 low-resolution copies of the high-resolution patches? → Isn’t the idea of the proposed KT method not to use patches due to their “local image context”? → How would then the volume-based benefits be covered in such a setup?


The authors properly answered every question and changed the manuscript accordingly. The authors have thought well of the idea to use their method in combination with the nnU-Net which seems plausible from their explanation and should be followed if an improvement in performance and novelty can be expected.

---

### Official Review · Reviewer_Ri82 · 2023-02-06

**Confidence:** 4
**Preliminary Rating:** 4
**Recommendation:** Poster

**Summary:**

The authors proposed to apply kaleidoscope transform (KT) for an input 3D image to 3D CNN. In this, the KT is applied on the input image and decomposes it into smaller downsampled images that are constructed as batches to be used for training. They demonstrate the usefulness of this transform on a small knee MR segmentation dataset and compare its performance with a context aggregation network.

**Strengths:**

1. It is a well written article and the methods and results section is easy to follow.
2. The experiments and presentation of results both quantitatively and qualitatively is well done.
3. The discussion of results is also good.
4. The KT transform is simple to use and can be used as a plug-and-play for different network architectures. It does help improve segmentation of some structures, especially bone.

**Weaknesses:**

1. The literature review is limited and the authors do not discuss many directions of relevant works in the literature that dealt with the problem of small annotated datasets like semi-supervised learning, data augmentation and self-supervised learning that deal with this topic.
2. The authors compare their method to only one previous work and do not provide comparisons with many previous works in the literature on the topics mentioned in the above comment that work with small datasets.
3. The authors do not use data augmentation in training for the methods in section 2.3 on page 3. It has been shown in many previous works that data augmentation helps improve segmentation results when one has limited data and is a kind of standard process in medical image segmentation. Can the authors explain why they chose to not apply any data augmentation here? I am not sure if the gains will be substantial provided we use correct data augmentation in training these different models?
4. Quantitative results:

    a) Results for smaller structures do not show much improvements with and without using KT. This is true also for the case with or without dropouts for many smaller structures like FC, TC, PC, MM, LM, PCL, ACL wrt to Dice score. Can the authors provide some rationale into why this behaviour is observed?

    b) Why are the HD metric scores bad for some structures like TB, TC, PB, ACL, PCL, etc for CAN3D (with dropout) model despite having a better or equivalent DSC score as compared to KAN3D (with dropout)? It is almost 2 times higher for the CAN3D model than the KAN3D model. Can the authors please explain this behaviour? Also, the HD metrics are in single digits (in mm) in the original CAN3D paper. Can the authors justify why they observed a big difference in HD scores between their implementation and original article - CAN3D?

5. The significance test:

    a) Can the authors explain how they compute the significance test? Is it done only on the 7 or 8 images present in the test set?

    b) Can the authors provide this value comparing CAN3D (with dropout) against KAN3D (with dropout)? Currently, this is not provided or it is not clear from the current presentation? It would be interesting to see if there are any significant gains between these two dropout models that perform well.


**Deanonymize Review:**

no

**Paper Type:**

validation/application paper

**Questions To Address In The Rebuttal:**

Can the authors please provide responses to the points (2-5) raised in the weakness section above.
The authors can focus on addressing the results section both quantitative results and significance test values and the corresponding discussions.

---

### Meta-Review · Area_Chair_LLbr · 2023-02-20

**Recommendation:** Accept (Poster)
**Confidence:** 5

**Metareview:**

**Positive:** The paper seems well-written, well-motivated and easy to follow. Their proposed approach seems to work well and the reviewers believe the paper will spark interest at MIDL.

**Negative:** Initially, there were several requests for clarification about the experiments and results section. However, the reviewers were satisfied with the revisions.

**Recommendation:** I thank the reviewers and authors for engaging in a constructive and interactive review process. Given that all reviewers agree, I recommend acceptance of the paper.